# Nutraceutical, Dietary, and Lifestyle Options for Prevention and Treatment of Ventricular Hypertrophy and Heart Failure

**DOI:** 10.3390/ijms22073321

**Published:** 2021-03-24

**Authors:** Mark F. McCarty

**Affiliations:** Catalytic Longevity Foundation, 811 B Nahant Ct., San Diego, CA 92109, USA; markfmccarty@gmail.com

**Keywords:** ventricular hypertrophy, congestive heart failure, nutraceuticals, nitric oxide, hydrogen sulfide, ER stress, FGF21, FGF23, marinobufagenin, plant-based diets

## Abstract

Although well documented drug therapies are available for the management of ventricular hypertrophy (VH) and heart failure (HF), most patients nonetheless experience a downhill course, and further therapeutic measures are needed. Nutraceutical, dietary, and lifestyle measures may have particular merit in this regard, as they are currently available, relatively safe and inexpensive, and can lend themselves to primary prevention as well. A consideration of the pathogenic mechanisms underlying the VH/HF syndrome suggests that measures which control oxidative and endoplasmic reticulum (ER) stress, that support effective nitric oxide and hydrogen sulfide bioactivity, that prevent a reduction in cardiomyocyte pH, and that boost the production of protective hormones, such as fibroblast growth factor 21 (FGF21), while suppressing fibroblast growth factor 23 (FGF23) and marinobufagenin, may have utility for preventing and controlling this syndrome. Agents considered in this essay include phycocyanobilin, N-acetylcysteine, lipoic acid, ferulic acid, zinc, selenium, ubiquinol, astaxanthin, melatonin, tauroursodeoxycholic acid, berberine, citrulline, high-dose folate, cocoa flavanols, hawthorn extract, dietary nitrate, high-dose biotin, soy isoflavones, taurine, carnitine, magnesium orotate, EPA-rich fish oil, glycine, and copper. The potential advantages of whole-food plant-based diets, moderation in salt intake, avoidance of phosphate additives, and regular exercise training and sauna sessions are also discussed. There should be considerable scope for the development of functional foods and supplements which make it more convenient and affordable for patients to consume complementary combinations of the agents discussed here. **Research Strategy:** Key word searching of PubMed was employed to locate the research papers whose findings are cited in this essay.

## 1. Sketch of the Pathogenesis of Ventricular Hypertrophy and Heart Failure

Ventricular hypertrophy (VH) progressing to heart failure (HF) is considered the leading cause of death in the U.S. today. This reflects the fact that such common cardiovascular disorders as coronary atherosclerosis, myocardial infarction, hypertension, chronic renal failure, diabetes, and obesity play a role in its induction. In this disorder, cardiomyocytes are subjected to chronically elevated oxidative stress, as well as a dysregulation of calcium control in which: calcium influx is inappropriately upregulated in some microenvironments. This intracellular calcium dysregulation leads to chronic activation of the key mediators calcineurin, nuclear factor of activated T cells (NFAT) and calcium/calmodulin-dependent protein kinase II (CaMKII), drivers of cardiomyocyte hypertrophy. It also compromises the swift systolic rise and rapid diastolic rise of intracellular free calcium, leading to decreased efficiency of contraction and relaxation, as well as an increased risk for arrhythmias. Additionally, this is associated with a depletion of calcium in the sarcoplasmic reticulum, giving rise to endoplasmic reticulum (ER) stress, a mediator of the cardiomyocyte apoptosis that contributes to progressive heart failure.

This pathological stew, which is self-reinforcing in many ways, is typically triggered by chronic mechanical strain—induced by uncontrolled hypertension, volume overload, valvular stenosis or failure, or loss of viable myocardium following myocardial infarctions or infections—and neurohormonal stimuli associated with the systemic stress reaction and/or kidney failure, such as adrenergic activity, angiotensin II, endothelin, and fibroblast growth factor 23 (FGF23). Other triggering stimuli include obesity/diabetes (associated with excessive exposure of cardiomyocytes to saturated free fatty acids), parathyroid hormone, and the hormone marinobufagenin (produced when salt-sensitive people eat salty diets). 

Most of these triggers either activate phospholipase C-β (PLC-β) via Gαq or boost cAMP production; FGF23 acts via FGFR4 to stimulate PLC-γ [1,2,3,4,5,6,7,8,9]. Mechanical stretch of cardiomyocytes opens pannexin-1 channels, allowing ATP to leak to the extracellular space, where it can activate P2Y purinoreceptors coupled to Gαq [10,11]. The centrality of Gαq/PLC-β to the VH/HF syndrome is revealed by the fact that transgenic mice bioengineered to overexpress Gαq in cardiomyocytes experience VH [12]. The diacylglycerol generated by activation of either PLC-β or PLC-γ plays a key mediating role in this syndrome; transgenic mice with cardio-specific overexpression of diacylglycerol kinase are resistant to hypertrophy when challenged with isoproterenol infusions or surgically induced pressure overload (transverse aortic constriction—TAC) [13]. The diacylglycerol generated by PLC activity increases the open probability of TRPC3/TRPC6 channels in caveolar microdomains; this influx somehow has a priming effect on calcium influx through neighboring L-type calcium channels [14,15,16,17,18]. (Generation of inositol-1,4,5-triphosphate via PLC activity has comparatively little impact on calcium release from sarcoplasmic reticulum in cardiomyocytes [2]) This local calcium increase activates the phosphatase calcineurin, likewise localized in caveolae, which can then dephosphorylate the transcription factor NFAT, enabling its migration to the nucleus, where it promotes transcription of genes coding for hypertrophic proteins and cardiac natriuretic peptides [16,19,20]. One of the effects of NFAT-driven transcription is an increase in TRPC6 expression, which in turn helps to maintain NFAT activation [21]. The increase in adrenergic activity usually associated with this syndrome exerts its effects via cAMP, which acts via the G-protein exchange factor EPAC to promote chronic activation of CaMKII; other factors promoting this activation include increased calcium influx within microenvironments, as well as oxidative stress, which induces calcium-independent activation of CaMKII via oxidation of methionine groups [22,23,24,25,26]. CaMKII promotes hypertrophy by phosphorylating and thereby inducing the nuclear export of class II histone deacetylases, which impede transcription of key hypertrophic genes [27,28,29]. CaMKII also promotes diastolic calcium leak from the sarcoplasmic reticulum by phosphorylating ryanodine receptors (RyR2) [30,31]. Additionally, CaMKII activity exerts transcriptional effects which perturb mitochondrial function, increasing mitochondrial inner membrane potential and boosting superoxide production at complex I [32]. Hence, chronic activation of CaMKII is a mediator of the oxidative stress typical of VH/HF, which in turn maintains the chronic activation of CaMKII in a vicious cycle. CaMKII may also amplify the ability of increased cytoplasmic calcium to induce oxidative stress and apoptosis by boosting activity of the uniporter which enables mitochondrial calcium influx—although this finding has been disputed [33,34]. 

cAMP also acts via protein kinase A (PKA) to modulate calcium dynamics—increasing the open probabilities of RyR2 and L-type calcium channels (thereby accounting for its positive inotropic effects), while up-regulating activity of the sarcoplasmic reticulum’s calcium pump SERCA2 via phosphorylation of the inhibitory protein phospholambam [35,36,37,38,39,40]. 

Cardiac mechanical strain also triggers the conversion of cardiac fibroblasts to myofibroblasts, which proliferate and secrete excessive amounts of collagen and other ground substance proteins, giving rise to the cardiac fibrosis and structural stiffness that contributes importantly to inefficient diastolic relaxation. This phenomenon is of key importance to the genesis of heart failure with preserved ejection fraction (HFpEJ), although it is usually also a feature of heart failure with reduced ejection fraction. When cardiomyocytes are subjected to mechanical strain, they synthesize and secrete increased amounts of transforming growth factor-β (TGFβ) and connective tissue growth factor (CTGF), which provide the paracrine stimulus to cardiac fibroblasts, converting them to myofibroblasts. The mechanism whereby mechanical strain induces increased secretion of TGFβ/CTGF has been partially defined: Strain in cardiomyocytes opens pannexin-1 channels that release ATP to the extracellular space, where it stimulates P2Y6 receptors to activate Gα12/13 G proteins; these in turn trigger activation of the G protein RhoA and its target kinase ROCK, which induce increased synthesis and secretion of TGFβ/CTGF [10,41,42,43]. Angiotensin II signaling in cardiomyocytes likewise can activate Gα12/13 [44]. TGFβ signaling within cardiac fibroblasts requires hydrogen peroxide production by Nox4, and is suppressed by nitric oxide/cGMP/protein kinase G (PKG) activity [45,46,47,48]. PKG-mediated phosphorylation of Smads opposes TGFβ-induced activation of Smad signaling; PKG also acts upstream in the fibrotic process by inhibiting RhoA activation [48,49,50]. Additionally, activation of ER stress plays a role in driving conversion of TGFβ-stimulated fibroblasts to myofibroblasts [51,52,53,54]. 

Cardiac fibrosis and hypertrophy of cardiomyocytes is commonly associated with capillary rarefication, which promotes regional hypoxia and increases arrhythmic risk; this reflects a failure of angiogenic mechanisms to respond appropriately to the increase in cardiac mass. Although cardiac vascular endothelial growth factor (VEGF) expression is enhanced initially after a chronic increase in cardiac afterload, VEGF levels eventually fall if this overload is maintained, leading to capillary rarefication, cardiac dilatation, and heart failure [55,56]. 

Loss of nitric oxide synthase (NOS) activity, often reflecting oxidant-mediated uncoupling, is another key feature of the VH/HF syndrome; this loss makes an especially key contribution to the pathogenesis of HFpEF, in which inefficient diastolic relaxation is the key problem [57,58]. Cardiomyocytes express NOS1 and NOS3 constitutively, whereas coronary vascular endothelium expresses NOS3; NOS2 can be induced in the heart via certain pro-inflammatory hormones and bacterial products [59,60]. Most of the protective effects of NO in the heart appear to be mediated by PKG. PKG promotes optimal activity of the SERCA2A calcium pump by phosphorylating its inhibitor phospholamban (as PKA does) [39]. This boost in SERCA2A activity improves diastolic relaxation by inducing a rapid diastolic decline in intracellular free calcium; this also helps prevent ER stress [61]. PKG-mediated phosphorylation of titin and tubulin in the contractile apparatus improve the efficiency of diastolic relaxation (a “lusitropic” effect); this is a key reason why loss of NO bioactivity promotes HFpEF [62,63,64]. PKG also helps prevent microenvironmental increases in free calcium by conferring an inhibitory phosphorylation on TRPC3/TRPC6 and L-type calcium channels [65,66,67]. PKG also suppresses the activity of the myocardial Na+/H+ exchanger (NHE-1); this exchanger promotes influx of extracellular calcium indirectly by boosting Na influx, which in turn drives calcium influx via the Na+/Ca+2 exchanger [68]. PKG, via phosphorylation of the G protein RhoA, blunts the ability of mechanical strain to promote cardiomyocyte synthesis and release of TGFβ and connective tissue growth factor (CTGF), which act in a paracrine manner on cardiac fibroblasts to trigger cardiac fibrosis; loss of effective NO bioactivity thus makes a key contribution to the pathogenesis of HFpEF, both by promotion of cardiac fibrosis and by loss of lusitropic activity [49,50,69]. Additionally, NO is a mediator of the angiogenic response that is needed when the heart tissue hypertrophies; hence, loss of efficient NO production is a cause of capillary rarefaction in VH/HF [70,71,72]. Finally, efficient NOS3 (“endothelial” NOS) activity in the peripheral vasculature helps control afterload by mediating endothelium-dependent flow-mediated vasodilation—which also aids exercise capacity in heart failure—and moderating systemic blood pressure [73]. Perversely, flow-mediated dilation tends to be impaired in heart failure owing the reduction in pulsatile shear associated with decreased cardiac output; episodic increases in shear stress associated with exercise training or regular sauna sessions help to correct this problem [73,74,75]. For all of these reasons, measures which support efficient NOS activity can be beneficial in VH/HF. 

The biological gas hydrogen sulfide (H_2_S) likewise plays a protective role in this syndrome [76,77]. Studies with rodent and cell culture models of VH/HF reveal that H_2_S can act to oppose cardiac hypertrophy, fibrosis, and functional impairment, while promoting cardiac angiogenesis [76,77,78,79,80]. The likelihood that these effects are of physiological significance is supported by reports that plasma H_2_S levels are reduced relative to controls in HF patients and in mice in whom HF has been induced by transverse aortic constriction (TAC); myocardial H_2_S is also low in mice subjected to TAC [76,81]. Moreover, mice genetically lacking the H_2_S-synthesizing enzyme cystathionine-gamma lyase show an exaggerated response to HF induction via TAC, whereas the opposite is true of transgenic mice with cardiospecific overexpression of this enzyme [81]. Studies attempting to define the direct targets of H_2_S in mediating this protection have reported that H_2_S administration increases activation of eNOS via Akt, increases cardiac expression of antioxidant enzymes via Nrf2 induction, and boosts cardiac expression of vascular endothelial growth factor [77,81]. Nrf2—nuclear factor erythroid 2-related factor 2—is a transcription factor that, by binding to antioxidant response elements in the promoters of a number of genes, promotes their transcription; these genes code for an array of enzymes that aid control of oxidative stress and that metabolically transform carcinogens and other toxins to hydrophilic derivatives that can be excreted. One of the enzymes induced is rate-limiting for the synthesis of the ubiquitous cellular antioxidant glutathione. This phenomenon is known as “phase 2 induction” [82]. 

Increased eNOS activation via Akt-mediated phosphorylation (eNOS expression is not altered) may be of importance to H_2_S’s protective activity in VH/HF, as eNOS knockout mice subjected to TAC are not benefited by H_2_S therapy [81]. Akt activation may also mediate the up-regulation of Nrf2 activity induced by H_2_S; Akt inhibits GSK-3β activity, which phosphorylates and antagonizes nuclear translocation of Nrf2 [83]. Additionally, Akt activation would be expected to oppose cardiomyocyte apoptosis [84].

Current pharmaceutical measures for VH/HF include beta-blockers, angiotensin II antagonists, and antagonists of the mineralocorticoid receptor. The former two inhibit activities that, as noted, are key triggers for this syndrome. Cardiac mineralocorticoid receptors (MR), which are constitutively active owing to cortisol binding, contribute to hypertrophy and fibrosis in ways that are still somewhat obscure; non-genomic activation of Nox2-dependent NADPH oxidase activity may play a key role in this regard [85,86,87]. Pharmaceutical control of hypertension and volume overload, as well as surgical correction of valvular dysfunction, are often employed as ancillary measures. Despite the well-established utility of these measures, VH usually worsens in most patients over time and progresses to HF, itself progressive. 

In the discussion which follows, we examine nutraceutical, dietary and lifestyle strategies which may have the potential to complement standard therapy in the prevention and control of the VH/HF syndrome. These measures are intended to: attenuate oxidative stress; suppress ER stress; support efficient production of NO and H_2_S; and mimic the bioactivity of NO. Other measures which appear likely to be beneficial include: optimizing magnesium (Mg) status; supplementing with taurine, carnitine, copper, and orotate; a whole-food plant-based diet, low in saturated fats, moderate in salt, and rich in potassium; regular exercise training; and sauna.

## 2. Attenuating Oxidative Stress

Hydrogen peroxide, by interacting with relatively acidic cysteines as well as methionines in key proteins, perturbs the function of many proteins in a way that promotes the VH/HF syndrome [88,89]. In particular, oxidation of RyR2 sensitizes it to calcium-mediated opening and promotes diastolic calcium leak [90,91,92]; oxidation of SERCA2 compromises its pumping activity [93,94,95]; oxidation of TRPC3/TRPC6 enhances calcium influx through these channels and sensitizes them to activation by diacylglycerol [96,97]; and oxidation of methionine groups in CaMKII locks it in an activated conformation even after intracellular free calcium has declined [24,25,26]. Oxidation and subsequent S-glutathionylation of NOS induces its uncoupling [98,99]. Hydrogen peroxide stemming from Nox4 plays a mediating role in the TGFβ signaling within cardiac fibroblasts which drives cardiac fibrosis [45,46]. Superoxide and its downstream products peroxynitrite and hydroxyl radical likely also contribute to the adverse impact of oxidative stress in VH/HF; in particular, peroxynitrite promotes uncoupling of NOS (thereby suppressing NO production while boosting that of superoxide) by oxidizing tetrahydrobiopterin [100,101]. Moreover, peroxynitrite further compromises NO signaling by inducing an inhibitory oxidation of soluble guanylate cyclase [102]. Oxidant stress also promotes NOS uncoupling by impairing the activity of dimethylarginine dimethylaminohydrolase (DDAH), which catabolizes the endogenous NOS inhibitor/uncoupler asymmetric dimethylarginine (ADMA); DDAH is inhibited by covalent interaction with 4-hydroxynonenal, a breakdown product of peroxidized membrane lipids [103,104,105]. Additionally, superoxide can react spontaneously with NO, destroying its bioactivity and generating toxic peroxynitrite in the process [106]. 

The centrality of hydrogen peroxide excess in the VH/HF syndrome is highlighted by studies with mice bioengineered to overexpress either mitochondrial or peroxisomal catalase—the former targeting mitochondria, the latter the cytoplasm [94,95,107]. These mice prove to be substantially though not wholly protected from VH/HF induced by transaortic banding or isoproterenol infusion; they are also resistant to age-related cardiac dysfunction. These findings strongly emphasize the desirability of controlling oxidative stress in VH/HF. 

The main sources of superoxide/hydrogen peroxide in this syndrome appear to be mitochondria, NADPH oxidase complexes, and uncoupled NOS; activated xanthine oxidase may also play a subsidiary role in this regard, but allopurinol appears to provide little benefit in HF [108,109,110]. Whereas chronic activation of CaMKII is known to promote mitochondrial superoxide production at complex I, locally elevated cytoplasmic free calcium, when taken up by mitochondria, can likewise increase superoxide production at complex I by increasing mitochondrial inner membrane potential [32,111]. Cardiomyocytes express Nox2 and Nox4, whereas Nox2 is the primary form of NADPH oxidase in vascular endothelium [112]. Rodent studies employing the broad-spectrum NADPH oxidase inhibitor apocynin, or mice genetically engineered to lack Nox2 or Nox4, reveal that NADPH oxidase is a key contributor to the oxidative stress associated with VH/HF, and that suppression of NADPH oxidase activity substantially ameliorates this syndrome; likewise, activation of Nox2 in coronary endothelium leads to coronary endothelial dysfunction in VH/HF models [87,113,114,115,116,117,118,119,120]. Increased activity of Gαq/PLC-β, by generating diacylglycerol that can activate PKC activity, likely plays a role in promoting Nox2-dependent superoxide production in VH/HF; in addition, activated mineralocorticoid receptors are known to stimulate Nox2-dependent superoxide production in the heart via a non-genomic mechanism [85,86,87]. Nox4 is constitutively active, so increased activity of Nox4 in VH/HF likely reflects induction of this protein. Another key source of oxidative stress is uncoupled NOS, which in particular plays a role in diastolic dysfunction [57,59,60]. Measures for achieving recoupling of NOS—and hence controlling another source of cardiac oxidative stress—will be considered below.

Coenzyme Q_10_ (CoQ), which shuttles electrons from complex I and complex II to complex III in the electron transport chain of the mitochondrial inner membrane, was one of the first nutraceuticals to be employed with some success in the management of HF [121,122,123,124]; a recent meta-analysis of controlled clinical trials with this agent in HF concludes that CoQ therapy can reduce mortality and improve exercise capacity, though left ventricular ejection fraction (LVEF) did not improve [125]. Since complex I appears to be the chief source of mitochondrial superoxide in VH/HF, it is reasonable to suspect that CoQ’s therapeutic role in this regard reflects its ability to accept electrons from complex I, thereby relieving the over-accumulation of complex I electrons that promotes superoxide production; this might also be expected to aid electron flow down the chain and thereby improve efficiency of ATP generation. CoQ can also act directly as a scavenging antioxidant. CoQ is most efficiently absorbed in its reduced form ubiquinol [126,127,128].

The intracellular free bilirubin generated by heme oxygenase activity functions as a potent direct inhibitor of certain NADPH oxidase complexes, including those dependent on Nox2 and Nox4 [129,130,131,132,133,134]. Hence, oxidant-mediated heme oxygenase induction provides feedback control of oxidative stress [132]. Recent evidence that serum bilirubin correlates inversely with risk for LVH in hypertensives, independent of blood pressure, may reflect the ability of free bilirubin to blunt the contribution of Nox-mediated oxidative stress to the progression of VH [135,136]. Moreover, induction of heme oxygenase-1 activity opposes induction of VH in rats that are hypertensive or treated with angiotensin II [137,138]. Although bilirubin is too insoluble to be administered orally as an antioxidant nutraceutical, chemically related bilins in algae appear to have potential in this regard. The light-harvesting chromophore phycocyanobilin (PhyCB), a major component of certain algae and cyanobacteria such as spirulina, is a close chemical relative (and derivative of) bilirubin’s direct precursor biliverdin; within cells, it is rapidly reduced to phycocyanorubin, very similar in structure to bilirubin [139,140]. Indeed, orally or parenterally administered PhyCB shares free bilirubin’s ability to inhibit NADPH oxidase complexes—which likely rationalizes the profound and versatile protective effects of orally administered spirulina (or of phycocyanin, the protein which contains PhyCB as a covalently linked chromophore) in rodent models of health disorders driven by oxidative stress [140,141,142,143,144,145]. While the utility of oral PhyCB (or of oral spirulina or phycocyanin) has not yet been tested in rodent models of VH/HF, oral spirulina has been shown to be protective in doxorubicin-induced heart failure in mice [146]. *In vitro*, phycocyanin protects cardiomyocytes from this drug [147]. 

Reduced glutathione works with glutaredoxin to reverse the oxidant effects of hydrogen peroxide on cysteine groups; hence, measures which boost cardiac levels of free glutathione may be helpful in VH/HF [148,149,150]. This can be achieved with supplemental intakes of N-acetylcysteine (NAC), a delivery form for cysteine, the rate-limiting substrate for glutathione synthesis [151,152]. The efficiency of tissue glutathione synthesis declines in the elderly, owing both to reduced efficiency of the Nrf2-dependent induction of gamma-glutamylcysteine sythetase (the rate-limiting enzyme for glutathione synthesis), and a decline in tissue levels of cysteine [153,154,155]. NAC supplementation in the elderly has been shown to restore youthful tissue levels of glutathione [156]. Not surprisingly, supplemental NAC has been shown to be protective in rodent models of VH/HF [157,158,159]. Moreover, NAC supplementation also has the potential to enhance cardiac production of H_2_S, as cysteine is the primary substrate for its synthesis [160]. Potentially, phase 2-inducer nutraceuticals such as lipoic acid or ferulic acid could be used to complement NAC’s efficacy for boosting glutathione synthesis, as these agents promote Nrf2-mediated induction of gamma-glutamylcysteine synthetase [153,161]. They also boost induction of a range of other antioxidant enzymes which potentially could be protective in VH/HF—including heme oxygenase-1, which generates free bilirubin [162,163,164,165,166].

Oxidant-mediated oxidation of methionine groups in CaMKII plays a key role in promoting chronic activation of this enzyme. The enzyme methionine sulfoxide reductase-A (MSR-A) functions to restore these methionines to their proper conformation [24]. Zinc supplementation of elderly human subjects has been shown to boost expression of MSR-A in peripheral blood leukocytes; whether this effect also obtains in cardiomyocytes has not yet been studied [167]. Zinc supplementation also promotes induction of the cysteine-rich antioxidant protein metallothionein, which has been shown to be protective in rodent models of diabetic cardiomyopathy [168,169,170,171,172,173,174]. Metallothionine induction can antagonize the pro-oxidant activity of cadmium (Cd), a heavy metal contaminant that has been linked to increased risk for heart failure and other cardiovascular pathologies [175,176,177]. Indeed, a number of studies have linked increased cadmium body burden (as assessed by urinary Cd corrected for creatinine) to increased risk for and mortality from heart failure [178,179,180,181,182,183]. The utility of supplemental zinc in rodent models of VH/HF merits study. Curiously, in the AREDS1 study examining the potential of nutraceuticals for control of macular degeneration, supplementation with zinc (80 mg daily, complemented by 2 mg copper) was associated with a highly significant 27% reduction in total mortality over about 6 years of follow-up [184]. If this effect is confirmable, a favorable impact of supplemental zinc on the heart may be in part responsible for this protection. Indeed, a recent cross-sectional study has found that serum zinc levels are lower in patients with LVH than in healthy controls, and that serum zinc correlates inversely with left ventricular mass [185]. 

Selenium is an essential cofactor for various antioxidant enzymes that likely would be protective in VH/HF, including glutathione peroxidase, thioredoxin reductase, and methionine sulfoxide reductase-B1 [186,187]. Although dietary selenium intakes throughout most of the US are sufficient to optimize selenium’s availability for enzyme production, this is not true in certain regions of Europe and Asia where soil selenium is quite low. Hence, modest supplemental intakes of selenium might be beneficial for heart health in these regions [188,189]. Not surprisingly, severe pediatric selenium deficiency, as was seen in certain regions of China, was associated with a severe cardiomyopathy (Keshan disease) [190]. In a double-blind supplementation trial, in which elderly Swedes were supplemented with selenium + coenzyme Q or matching placebo for four years, the cardiovascular mortality rate was about half as high (HR: 0.51; 95%CI 0.36–0.74; *p* = 0.0003) in the active treatment group, and plasma levels of N-terminal brain natriuretic peptide (BNP) were significantly lower than in controls, suggestive of a favorable effect on VH; benefit was seen primarily in those with low baseline selenium, common in elderly Swedes [191,192,193].

Excess oxidant production in cardiomyocyte mitochondria, a feature of VH/HF, might be expected to lead to oxidant damage to the mitochondrial respiratory chain which in turn boosts superoxide production or compromises efficient ATP synthesis. Therefore, a lipid soluble membrane antioxidant such as astaxanthin—shown to be protective in rodent models of ischemia-reperfusion damage [194,195,196]—might be expected to aid preservation of mitochondrial function and minimize mitochondrial oxidant production in VH/HF. Indeed, administration of astaxanthin has been shown to lessen heart oxidant stress in aged rats given repeated injections of isoproterenol [197].

The neurohormone melatonin amplifies phase 2 induction by increasing expression of Nrf2 at the transcriptional level, likely by increasing expression of the clock protein Bmal1, a clock protein transcription factor which binds to the promoter of the Nrf2 gene [198,199]. However, melatonin, likely via Bmal1, also promotes transcription of the gene coding for the deacetylase Sirt1 [200,201]. Sirt1 exerts anti-fibrotic activity in the heart and other organs by opposing TGFβ signaling in several complementary ways. Sirt1 activity promotes proteasomal degradation of p300, a histone acetyltransferase which acts as a cofactor for TGFβ-induced Smad2/Smad3-mediated transcription [202,203,204]. Sirt1 also deacetylates Smads 2, 3 and 4, and some evidence suggests that this may also interfere with TGFβ-mediated signaling [205,206]. Melatonin administration has been shown to inhibit cardiac fibrosis in several rodent models of this condition [207,208,209,210,211].

## 3. Inhibiting ER Stress

As noted, the loss of sarcoplasmic reticulum calcium associated with VH/HF gives rise to ER stress [212]. This can lead to cardiomyocyte apoptosis via induction of CHOP and activation of JNK/p38 MAP kinase; ER stress also promotes cardiomyocyte inflammation via activation of NF-kappaB [213]. ER stress within fibroblasts promotes cardiac fibrosis [51,53]. The administration of certain chemical chaperones, such as tauroursodeoxycholic acid (TUDCA) and 4-phenylbutyrate, aids control of ER stress by promoting proper protein folding, and indeed has been shown to be highly protective in rodent models of VH/HF [53,54,214,215,216,217,218]. TUDCA is of particular interest in this regard because it is a naturally occurring bile salt that, while used clinically to treat certain cholestatic liver disorders, is also available as a nutraceutical. The oral doses of TUDCA found to be protective in rodent models of VH/HF extrapolate to about 2–3 g daily (preferably in divided doses) in humans. Such a regimen would likely be feasible, though rather expensive—4–6 US dollars daily at current retail prices.

Berberine, a phytochemical found in certain Chinese herbs, has long been used as a treatment for diabetes in China [219,220,221]. Its efficacy in this regard is thought to reflect its ability to activate AMPK in a manner comparable to the diabetes drug metformin [222,223]. However, berberine has also been used in China as a treatment for heart failure. Controlled trials have demonstrated that berberine administration at 1–2 g daily can increase ejection fraction, increase exercise capacity, lessen ventricular premature contractions, and decrease mortality in heart failure patients; results in rodent models of VH/HF reinforce this impression [224,225,226,227]. Curiously, there is also some evidence that metformin is beneficial in diabetics with heart failure; these findings thus point to the possibility that activation of AMPK is protective in this syndrome [228,229]. If so, improvement of ER stress could be a key mediator of this benefit. 

A number of rodent and cell culture studies demonstrate that activation of AMPK opposes ER stress [230,231,232,233,234]. AMPK functions as a detector of energy shortage; it promotes adaptation to such a shortage by boosting the activity of pathways that generate ATP, while slowing those that consume it [235]. One of the ways it accomplishes this is to confer an activating phosphorylation on eEF2 kinase, which in turn phosphorylates eEF2 [236,237]. The latter is an elongation factor that promotes swift translation of mRNAs; its activity in this regard is blunted by the phosphorylation mediated by eEF2 kinase. Hence, AMPK conserves ATP by slowing the rate of protein synthesis globally. Since ER stress reflects an inability of ER protein folding mechanisms to keep up with the rate of protein synthesis, the slowing of protein synthesis induced by AMPK activity could be expected to lessen ER stress, as in fact is observed experimentally. Furthermore, this mechanism would likely be complementary to that of TUDCA, which aids the efficiency of proper protein folding. Hence, berberine + TUDCA (abetted by measures for control of oxidative stress, as described above), might be expected to have general utility for combatting ER stress. In addition, it should be noted that AMPK can also confer an activating phosphorylation on eNOS [238,239]—and NO/cGMP/PKG oppose ER stress by inducing phosphorylation of phospholamban and hence activating SERCA2a.

## 4. Supporting NO/cGMP Production

The antioxidant measures cited above would likely help to prevent NOS uncoupling by decreasing peroxynitrite-mediated oxidation of tetrahydrobiopterin to dihydrobiopterin, and oxidant-mediated inhibition of DDAH. NOS is only fully coupled when it binds to both arginine and tetrahydrobiopterin; ADMA and dihydrobiopterin compete with these compounds for binding to NOS, and hence promote uncoupling [240]. Supraphysiological doses of folic acid have been shown to induce increased expression of dihydrofolate reductase in the vascular endothelium [241,242,243]; this enzyme is capable of reducing dihydrobiopterin back to its tetrahydro- cofactor form. Additionally, within cells, folate is converted to reduced forms which can efficiently scavenge peroxynitrite-derived radicals, and thereby protect tetrahydrobiopterin [244,245,246]. Hence, high-dose supplemental folate has been found to promote proper coupling of eNOS in vascular endothelium and improve dysfunctional endothelium-dependent vasodilation [247,248,249]. Whether high-dose folate could also induce dihydrofolate reductase in cardiomyocytes has not been studied; cardiomyocyte expression of this enzyme tends to be low [250]. Indeed, that’s why direct administration of tetrahydrobiopterin or its precursor sepiapterin has proved of minimal clinical value for recoupling NOS in cardiomyocytes—tetrahydrobiopterin is rapidly oxidized to its dihydro- form, which tends to accumulate and inhibit access of tetrahydrobiopterin to the enzyme. In any case, high-dose folate likely would promote coupling of eNOS in the coronary vascular tree; indeed, acute oral administration of 30 mg folate has been shown to improve flow-mediated vasodilation in the coronary tree of angina patients [251]. In mice subjected to transverse aortic constriction, pre-feeding with a diet supplemented with high doses of both folate and cobalamin was associated with a reduction in LVH and a preservation of EF; however, this benefit did not reflect control of oxidative stress, but rather a preservation of mitochondrial biogenesis likely attributable to increased methylation reactions [252]. Folate has also been shown to be highly protective for doxorubicin-induced cardiomyopathy in mice, as well as in a rat model of myocardial ischemia-reperfusion injury [253,254].

HF is associated with increased plasma ADMA levels—in part owing to decreased renal perfusion—and elevated ADMA is associated with poor prognosis [255,256,257,258,259,260]. Additionally, independent of renal function, ADMA levels correlate with left ventricular mass; this correlation has been observed in the normal aging population [261,262,263,264]. These findings suggest that, whereas diminished cardiac output can cause elevations of ADMA, elevations of ADMA may also increase risk for VH/HF. When ADMA levels are elevated—as they often are in patients with VH/HF—boosting tissue levels of NOS’ substrate arginine can promote NOS coupling in the heart by increasing the arginine/ADMA ratio [240,265]. Counterintuitively, supplementation with citrulline (remarkably, a major component of watermelon juice [266]) is more efficient for increasing tissue levels of arginine than is supplementation with comparable doses of arginine; this reflects the fact that inducible arginase activities in gut bacteria and the liver tend to degrade orally administered arginine to ornithine before it can reach the systemic circulation [267,268,269]. Citrulline, in contrast, is not degraded by arginase (indeed, it competitively inhibits this enzyme), and, in cells expressing NOS activity, it is rapidly converted to arginine. Furthermore, citrulline protects this derived arginine within these tissues by inhibiting arginase. Moreover, citrulline, as contrasted to arginine, has a pleasant mild flavor and is readily administered in multi-gram doses in fluids [240]. Hence, supplemental citrulline may have potential for use in management of VH/HF. Indeed, in a randomized controlled study (lacking a placebo), patients with HF receiving 3 g citrulline daily for 4 months achieved improvements in LVEF, RVEF, and endothelial function that were significant both with respect to baseline and controls [270]. In 35% of the treated patients, functional class improved.

Premenopausal women, as compared to men, are less prone to develop VH/HF, and have a better prognosis when they do; this protection is lost after menopause [271,272,273]. Conversely, postmenopausal estrogen replacement therapy has been shown to oppose increases in LV mass [274,275,276]. Likewise, estrogen administration protects ovariectomized female rodents in models of VH induction, and, importantly, male rats are also protected [277,278,279,280]. Cardiomyocytes express both α and β forms of the estrogen receptor, and this expression does not differ by sex [281,282]. Studies with estrogen receptor knockout mice reveal that estrogen receptor-β (ERβ), rather than estrogen receptor-α (ERα), mediates protection from VH [283,284,285]. Consistent with this finding, pharmaceutical agonists specific for ERβ protect rodents from VH, and cardiomyocyte-specific overexpression of ERβ is likewise protective [286,287,288]. Increased cardiac expression of eNOS appears to be at least partially responsible for this ERβ-mediated protection [289,290].

Fortuitously, the soy isoflavone genistein, when administered in doses that could feasibly be achieved with a high intake of soyfoods, produces plasma levels of free unbound genistein that are sufficient to activate ERβ, but too low to meaningfully activate ERα [291,292,293]. The daidzein metabolite S-equol—produced by intestinal bacteria to varying extents after ingestion of soy isoflavones—likewise can selectively activate ERβ in concentrations produced by soyfood ingestion [294]. This is presumably why soy isoflavones can provide a range of health benefits without provoking feminizing side effects in men [293]. This also rationalizes a number of recent reports that oral genistein is beneficial in rodent models of VH/HF [295,296,297,298,299,300]. The protection afforded by genistein in these studies is associated with increased myocardial activity of eNOS, and is abolished by the NOS inhibitor L-NAME, but not by a specific inhibitor of inducible NOS [295,297,300]. This increased activity of eNOS appears to reflect both increased expression of the enzyme and increased Akt-mediated phosphorylation; studies differ as to the relative importance of these effects. Asian epidemiology evaluating the impact of soyfood consumption on risk for this syndrome unfortunately is not yet available.

Raw cocoa powder is rich in the flavanols epicatechin, catechin, and polymerized complexes thereof, known as oligomeric procyanidins (OPCs). The favorable impact of cocoa on cardiovascular health has been traced to the ability of cocoa flavanols, notably epicatechin and some OPCs, to up-regulate the eNOS activity of vascular endothelium, boosting endothelium-dependent vasodilation [301,302]. Although this mechanism may not increase eNOS activity in cardiomyocytes and fibroblasts, it can increase the heart’s exposure to NO produced by the coronary microvasculature. Moreover, by improving flow-mediated vasodilation systemically, it can decrease the ventricular afterload that the heart must cope with. In vitro studies show that both epicatechin and high-molecular-weight OPCs can act on endothelial cells to promote Ser1177 phosphorylation of eNOS by activating the PI3K/Akt pathway; this boosts eNOS activity and up-regulates its further activation by stimuli such as shear stress which increase cytosolic free calcium in endothelial cells [303,304]. A recent crossover clinical study has found that ingestion of about a gram of cocoa flavanols daily by HF patients can decrease both diastolic blood pressure and N-terminal-pro-BNP levels [305]. Moreover, other food and herbal extracts rich in OPCs have potential for improving defective endothelial function in HF patients [306]. OPC-rich extracts of hawthorn (Crataegus oxyacantha) have long been employed in the clinical management of HF, and their high-molecular weight OPCs have likewise been found to activate eNOS in vitro via PI3K/Akt [307,308]. Although results of clinical studies with hawthorn extracts have yielded inconsistent results, a meta-analysis of randomized controlled trials has concluded that treatment with standardized OPC-rich hawthorn extracts can indeed confer symptomatic benefit in HF, increasing exercise capacity and lessening shortness of breath and fatigue [309]. 

Dietary nitrate is reduced to nitrite by oral bacteria, and this nitrite can be absorbed. In tissues that are relatively hypoxic and/or acidic, certain deoxygenated heme proteins such as hemoglobin, myoglobin, and xanthine oxidoreductase are capable of reducing nitrite to NO [310,311]. Hence, dietary nitrate may represent a NOS-independent strategy for increasing NO production in ischemic tissues, thereby improving their perfusion. Clinical studies with supplemental nitrate, often administered as beet juice, document a modest reduction in systemic blood pressure—of meaningful magnitude in hypertensives—that might be worthwhile for some patients with VH/HF [312,313,314,315,316]. In addition, nitrate may have an effect on exercise efficiency that could be of particular value to HF patients, whose exercise capacity is notably impaired. For reasons that remain mysterious, nitrate administration has been found to modestly decrease the oxygen uptake required to achieve a given power output in skeletal muscle; this might be expected to improve performance in certain types of exercise when muscle perfusion is diminished—as when cardiac output is low [317,318]. Moreover, in rats with HF owing to a previous induced infarction, nitrate administration for 5 days prior to an exercise trial was found to boost skeletal muscle blood flow by about 20% [319]. In HF patients, beet juice administration has been reported to modestly increase submaximal endurance and peak power production [320,321]. Three clinical trials, including one specifically targeting patients with HF with preserved ejection fraction, have shown exercise benefits with beet juice ingestion; a study enrolling only patients with reduced ejection fraction did not observe benefit [320,321,322,323]. Whether dietary nitrate could boost NO production in relatively hypoxic myocardial regions, thereby enhancing cardiac cGMP generation and opposing progression of VH/HF, has not yet been assessed, either clinically or in rodents. Hence, aside from the impact of its modest anti-hypertensive activity, it is not yet clear whether dietary nitrate can provide more than symptomatic benefit in HF. 

Drugs that directly activate soluble guanylate cyclase (sGC)—in its native or oxidatively inhibited form—have shown marked utility in rodent models of VH/HF, and are now being studied clinically in HFpEF [324,325,326,327,328,329]. Largely overlooked is the fact that, in concentrations two orders of magnitude higher than the physiological range, the B vitamin biotin can boost sGC’s production of cGMP by 2–3-fold [330,331,332]. (Whether it shares the ability of sGC stimulator drugs to amplify this enzyme’s sensitivity to NO has not been studied.) In spontaneously hypertensive stroke-prone rats, high oral doses of biotin have been shown to exert an antihypertensive effect that is abrogated by an sGC inhibitor; this supplementation also led to a marked reduction in stroke-mediated mortality [333]. More recently, high-dose biotin has shown well-tolerated clinical utility in progressive multiple sclerosis, an effect which might be mediated by CNS cGMP [334,335]. Benefits of high-dose biotin in diabetic rodents and humans are also likely traceable to increased cGMP production [336]. In light of the fact that high-dose biotin appears to be safe, well-tolerated (reflecting the modest extent to which it can activate sGC, as compared to NO), and reasonably affordable in doses likely to achieve mild systemic activation of sGC, its utility in VH/HF merits evaluation. However, it should be noted that use of high-dose biotin can interfere with certain clinical assays that use streptavidin-biotin technology, mandating its temporary discontinuation in some circumstances [337].

Oxidation of the ferrous heme of sGC—which is readily induced by peroxynitrite—renders it unstable, such that the heme can dissociate from the enzyme, negating the ability of NO, sGC stimulator drugs, and likely biotin to activate it [102]. Hydrogen sulfide, more effectively than glutathione, functions to maintain the heme iron of sGC in its reduced ferrous valence, thereby stabilizing the enzyme [338]. Hence, measures which promote H_2_S synthesis in the heart may complement the utility of biotin or NO for boosting cGMP production. 

Phosphodiesterases targeting cGMP have evident potential for managing VH/HF. PDE5 inhibitors, employed clinically to treat erectile dysfunction, are useful in rodent models of VH/HF [61,339]. Human trials demonstrate that the PDE5 inhibitor sildenafil provides hemodynamic, symptomatic, and mortality benefits in HF with reduced ejection fraction, but not in HFpEJ [340,341,342]. The basis of this discrepancy is not clear—perhaps it reflects a very severe loss of NO bioactivity in the latter syndrome. Recently, H_2_S has been shown to have PDE5-inhibitory activity; hence H_2_S acts within the heart both to promote the synthesis of NO and support its bioactivity [343].

## 5. Optimizing Omega-3 Status 

Long-chain omega-3 fatty acids, notably the eicosapentaenoic acid (EPA) and docosahexaenoic acid (DHA) found in fish oil, work in a variety of ways to protect the heart and cardiovascular system [344]. The impact of omega-3 status on risk for sudden-death arrhythmias has been traced to the ability of membrane omega-3s to interact with cardiac ion channels—notably fast voltage-dependent sodium channels and L-type calcium channels—in a way that reduces their open probability, rendering cardiac membranes more electrically stable [345,346]. Diets rich in EPA have been shown to decrease cardiac fibrosis in rodent models of pressure overload, reflecting the ability of EPA to activate GPR120 receptors on cardiac fibroblasts; this effect is dependent on an increase in fibroblast cGMP/PKG activity [347,348,349]. Additionally, the ratio of EPA to arachidonic acid (AA) in membranes or in plasma free fatty acids can determine the extent to which AA is converted to pro-inflammatory eicosanoid signaling factors [344,350]. EPA competes with AA for access to cyclooxygenase and other enzymes which produce such mediators; furthermore, these enzymes tend to convert EPA to mediators that are less pro-inflammatory/pro-aggregatory than those derived from AA, or that antagonize the interaction of AA-derived eicosanoids with their receptors. Importantly, the prostacyclin derived from EPA, PGI3, is as effective for promoting human platelet stabilization as is the PGI2 derived from AA—whereas the thromboxane derived from EPA is inactive [351,352,353]. A recent prospective study in patients with heart failure found that those in whom the EPA/AA ratio of plasma lipids was relatively high experienced notably lower subsequent mortality than those in whom this ratio was lower; curiously, this effect was more dramatic in patients taking statins [354]. Other studies have found that this ratio predicts the stability of coronary plaque—a high ratio correlating with more stable plaque, likely pointing to a role for AA-derived eicosanoids in plaque destabilization; furthermore, a high EPA/AA ratio would tend to stabilize platelets by decreasing thromboxane activity [355,356,357,358]. Hence, the fact that statin-treated heart failure patients achieve greater protection from a high EPA/AA ratio may reflect the fact that statin therapy is acting as a marker for coronary atherosclerosis and hence risk for myocardial infarct.

Since patients with VH/HF are more prone to arrhythmias, and often have coronary plaque, and since sufficient levels of EPA appear to provide protection from cardiac fibrosis, it would seem reasonable for VH/HF patients to optimize their omega-3 status, particularly with respect to EPA. The utility of a given daily intake of EPA/DHA will be influenced by the amount and type of other lipids ingested [359]. If one’s diet is low in total fat, and if omega-6 fatty acids (capable of being converted to AA) constitute a modest proportion of that fat, a given intake of EPA will yield a higher fraction of EPA in membranes and plasma free fatty acids, as well as a higher ratio of EPA/AA. It is notable that plant-based diets are devoid of AA. Hence, a low-fat plant-based diet, in which oleic acid and α-linolenic acid predominate over linoleic acid, may be the ideal setting for gaining the most protective benefit from a given supplemental intake of EPA/DHA. The equivocal results of many clinical trials with supplemental omega-3s may reflect a modest dose interacting with high-fat, high omega-6 diets. 

## 6. Ancillary Supplements—Mg, Orotate, Carnitine, Taurine, Glycine, and Copper

Epidemiologically, low serum or dietary levels of magnesium (Mg) have been associated prospectively with increased risk for VH/HF [360,361,362,363,364,365,366]. Moreover, certain drugs typically used in management of heart failure can compromise Mg status [367]. Although long-term supplementation trials have not yet evaluated the impact of magnesium in clinical VH/HF, the possibility that maintaining relatively high cardiac levels of Mg may provide some protection in this syndrome merits evaluation. At physiological cellular concentrations, Mg competes with calcium for binding to the N-terminal arm of calmodulin [368,369,370]. When partially Mg-bound, this protein is less capable of activating certain enzymes than is calmodulin in which Ca occupies all the binding sites. Hence, when cytoplasmic levels of free Mg are relatively low, the ability of elevated Ca to activate calmodulin-dependent enzymes—such as calcineurin and CaMKII may be up-regulated. Additionally, the ability of Ca to open RyR2 receptors is opposed by Mg—so cytoplasmic levels of free Mg may regulate this interaction as well [371]. Assuring good Mg status may be smart policy in patients coping with VH/HF. Evidence that calcium supplementation may increase cardiovascular risk in women may reflect the ability of such supplementation to impair Mg absorption and retention in the context of low-Mg diets [372,373]. This would argue for complementing calcium supplementation with concurrent Mg supplementation.

Mg also helps to prevent medial vascular calcification by counteracting the tendency of high-normal phosphate exposure to induce a phenotypic transition of vascular smooth muscle to osteoblast-like cells [374,375,376]. The mechanism of this protective effect is unclear, but intracellular Mg uptake is required. In kidney failure patients on dialysis—in whom serum phosphate levels tend to be high—as well as in the general population, higher serum Mg levels predict decreased risk for vascular calcification [377,378,379,380,381]. Since arterial and valvular calcification can increase the afterload on the heart, good Mg status and other measures which help to prevent such calcification have some value for primary prevention of VH/HF. In this regard, it should be noted that adequate intakes of the bacterially synthesized menaquinone form of vitamin K may also have utility, by promoting optimal γ-glutamyl carboxylation of the matrix Gla protein, an antagonist of calcium crystal deposition [382,383,384]. Menaquinones achieve better distribution to peripheral tissues than does the plant-derived phylloquinone form of vitamin K, which is largely retained by the liver for manufacture of clotting proteins [385].

The Mg salt Mg orotate has shown utility in rodent models of VH/HF, and favorable effects have been reported with this agent in small, short-term clinical trials in heart failure [386,387,388,389]. One longer-term trial is of particular note: Mg orotate was tested versus placebo in a randomized controlled trial enrolling patients with severe HF; the active group received 6 g daily for a month, and 3 g daily for another 11 months. At the end of the trial, mortality had been 52% in the treated group vs. 76% in the placebo group (*p* < 0.05), and clinical symptoms had improved in 38% of the treated group [390]. As Mg per se seems unlikely to account for an effect of this magnitude, orotate was thought to make an important contribution in this regard. Orotate, a pyrimidine metabolite, is converted by the liver to beta-alanine, and returned to the circulation [391,392]. In heart, skeletal muscle, and the central nervous system, beta-alanine is a precursor for carnosine and other histidine-containing dipeptides which function both as antioxidants, membrane protectants, and as buffers for intracellular acidity [392,393]. Indeed, supplemental beta-alanine has been shown to improve performance in high-intensity exercise, likely by buffering the acidity stemming from lactic acid production [394,395]. A recent clinical trial has found that 500 mg carnosine daily improves walking time and quality of life in HF patients; this benefit likely was mediated by beta-alanine, since carnosine is rapidly degraded to its constituents after oral administration [396,397].

Contrary to recent concerns that dietary or supplemental carnitine might increase vascular risk by boosting trimethyl-N-oxide production, [398]. carnitine administration to patients who have experienced a myocardial infarct is associated with a marked reduction in overall mortality, as confirmed by meta-analysis [399,400,401]. Supplemental carnitine has also been studied as an adjunct to the management of HF in a number of controlled trials, mostly of modest size. A recent meta-analysis of these studies found that supplemental carnitine significantly improved ejection fraction and other cardiac functional parameters, and lowered plasma levels of cardiac natriuretic hormones [402]. A trend toward lower overall mortality in the carnitine-treated patients just failed to achieve traditional statistical significance (*p* = 0.06). The mechanistic basis of carnitine’s benefit in these patients remains unclear. In the setting of ischemic coronary disease, it is known that carnitine helps to limit lactic acid production in cardiomyocytes by disinhibiting pyruvate dehydrogenase (owing to increased conversion of acetyl-coA to acetyl-carnitine, and a consequent decline in the acetyl-coA/free coA ratio that regulates pyruvate dehydrogenase kinase activity [403]) and thereby diverting more pyruvate to oxidation rather than to lactate production; this effect might also be pertinent in cardiomyocytes that are relatively hypoxic owing to the capillary rarefication often associated with VH/HF [404]. Acid generation in cardiomyocytes tends to provoke calcium overload, since intracellular protons are exchanged for extracellular sodium, and the resulting increase in intracellular sodium acts to drive calcium uptake via the sodium-calcium exchanger [405]. Could acid-buffering be a physiological role of carnosine and other histidine-based small molecules synthesized in the heart [406]?

For reasons that remain unclear, multi-gram daily supplemental intakes of taurine exert a positive inotropic effect on the failing heart that appears to be safe (i.e., not associated with the increased risk for arrhythmias seen with cardiotonic glycosides); improvements in LV ejection fraction and walking distance have been reported in several controlled trials [407,408,409,410]. Similar benefits have been reported in rabbit models of heart failure [411,412]. Conversely, cats fed a taurine-deficient diet develop fatal cardiomyopathy (as they lack the capacity to synthesize this agent), and genetic knockout of the cardiac taurine transporter likewise causes heart failure [413,414]. Whereas taurine supplementation can provide functional benefit in pre-existing heart failure, whether optimal taurine status might provide protection with respect to progression of VH has received little study. A reason to suspect that it might has emerged recently. In vascular endothelium, taurine has been shown to promote increased expression of two enzymes that convert can convert cysteine to H_2_S—cystathionine-β-synthase, and cystathionine-γ-lyase [415]. Indeed, this effect appears to be a mediator of the anti-hypertensive and anti-atherosclerotic effects of taurine demonstrated in rodents [416,417,418]. Combined supplementation with NAC and taurine therefore appears to be a practical strategy for boosting the production of protective H_2_S in the vasculature [160]. Could taurine also increase expression of H_2_S-synthesizing enzymes in cardiomyocytes? This should be studied. In any case, taurine supplementation appears to be a worthwhile, safe, and inexpensive strategy for managing HF, though its mechanism of action at the molecular level still requires clarification. 

Glycine-gated chloride channels are expressed by a range of tissues, including cardiomyocytes [419]. Increases of plasma glycine within the physiological range increase the open probability of these channels, which exert a hyperpolarizing effect on cell types which do not concentrate chloride, as their activation induces an influx of chloride [420,421]. Indeed, glycine has been reported to hyperpolarize cardiomyocytes [419]. Hyperpolarization would be expected to down-regulate calcium influx through voltage-activated L-type calcium channels, which mediate much of the calcium influx that drives VH. Perhaps that rationalizes a recent report that a glycine-enriched diet attenuates the development of VH in mice subjected to pressure overload or angiotensin II administration [422]. In vitro, glycine blunted the ability of angiotensin II to evoke increased release of TGFβ from cardiomyocytes—pointing to a likely protective effect on cardiac fibrosis [422]. Glycine administration has also been shown to diminish infarct area in rats subjected to myocardial ischemia reperfusion [423]. Moreover, glycine stabilizes platelets in vitro, and bleeding time is increased in rats fed glycine [424]. Of considerable interest is a prospective epidemiological study in which higher serum glycine levels correlated with decreased risk for myocardial infarction in patients with stable angina; those in the fifth quintile of serum glycine, as compared to the first quintile, were about 30% less likely to develop an MI, after statistical adjustment for recognized risk factors [425]. Supplemental glycine may complement the utility of NAC administration for raising cellular glutathione levels [156]. Glycine has good potential for inclusion in functional foods and beverages, as it is highly soluble, mildly sweet, and inexpensive in multi-gram doses [421].

There is evidence that a decline of myocardial copper levels contributes to the failure of adaptive angiogenesis that leads to capillary rarefication as the overloaded heart transitions from hypertrophy to failure [56]. The basis of this decline requires clarification, although an increase in the production of copper-chelating homocysteine in heart tissue has been suggested as a basis for this [426]. In any case, copper is known to act as a co-factor in the HIF-1-driven transcription of the genes encoding VEGF and other proteins that promote adaptation to hypoxia. Copper does not influence HIF-1α stability or nuclear translocation, but is somehow necessary for binding of HIF-1 to its response elements in the promoters of its target genes; some evidence suggests that copper-mediated inhibition of the factor inhibiting HIF-1 (FIH-1) may be responsible for this effect [427]. Copper’s role in supporting angiogenesis has led to the exploration of copper deprivation as a strategy for slowing cancer growth [427]. Remarkably, correction of the decline in myocardial copper levels in rats subjected to pressure overload with a copper-enriched diet not only prevents a decline in VEGF levels and boosts myocardial angiogenesis, but also exerts an anti-hypertrophic effect [56]—possibly by lowering cardiac levels of VEGFR-2 [428]. Optimization of cardiac copper status may also have a favorable impact on the activities of copper-zinc-dependent superoxide dismutase and mitochondrial cytochrome oxidase in the failing heart [429]. Although clinical studies have not yet evaluated the impact of copper supplementation per se on VH/HF, a controlled trial in which HF patients received a multi-vitamin-mineral supplement or matching placebo observed a reduction in LV mass and an increase in LV ejection fraction—both modest, but significant—in those receiving the supplement [430]; this supplement provided 1.2 mg copper per day, and Klevay has suggested that this copper may have been a key mediator of the benefit observed in this supplementation trial [431]. Defective copper uptake by cardiomyocytes is also seen in diabetes, and diabetic cardiomyopathy in rats is remedied by a copper-specific chelating agent (trientine) which reverses this intracellular copper deficiency [432]. It should be noted that high-dose zinc supplementation has the potential to decrease body copper stores by inducing metallothionein in the intestinal mucosa—for which reason it has been used in the treatment of Wilson’s disease [433]; hence, if zinc supplementation is employed in the management of VH/HF, concurrent supplementation with copper may be particularly wise.

Primary hyperparathyroidism tends to be associated with increased ventricular mass—possibly reflecting a direct impact of parathyroid hormone (PTH) on cardiomyocytes—and treatment of this syndrome with parathyroidectomy tends to decrease ventricular mass [434,435]. It is therefore reasonable to suspect that the secondary hyperparathyroidism seen when vitamin D status is low might likewise increase VH risk. Indeed, epidemiological studies have found that low plasma levels of 25-hydroxyvitamin D (25-OHD) correlate inversely with ventricular mass, predict future onset of VH, and are associated with poor prognosis in pre-existing heart failure [436,437,438,439,440,441]. Nonetheless, supplementation trials with vitamin D in patients with VH or HF have yielded equivocal results, and a recent meta-analysis of controlled studies of vitamin D supplementation in HF patients found that such supplementation did not improve LV ejection fraction, decrease plasma N-terminal pro-B type natriuretic peptide, or increase 5-min walking distance [442]. It therefore seems likely that, when lower 25-OHD is found to correlate inversely with VH/HF risk, lower 25-OHD is simply functioning as a marker for metabolic syndrome/obesity, which themselves promote VH [443,444,445]. Indeed, Mendelian randomization studies focused on genetic determinants of plasma 25-OHD have concluded that, whereas low vitamin D status may indeed be a mediator of increased cancer and overall mortality, it does not appear to influence cardiovascular risk [446]. Mendelian randomization has not yet been employed to assess the impact of vitamin D status on risk for VH/HF per se, but it seems unlikely that vitamin D would fail to influence overall cardiovascular mortality if it had an important effect on risk for VH/HF. Hence, whereas vitamin D supplementation may prove to be wise from the standpoint of overall health, whether it will notably influence risk for VH/HF is dubious. Perhaps commonly occurring degrees of vitamin D deficiency do not elevate PTH sufficiently to have an important effect on the heart.

## 7. Protective Diet and Lifestyle Measures

Metabolic syndrome, central obesity, and diabetes substantially increase risk for VH/HF, independent of their common association with hypertension [447,448,449,450,451,452,453,454,455,456,457]. These disorders are characterized by high tissue exposure to free fatty acids, and there is reason to suspect that increased exposure to long-chain saturated fatty acids is a mediator of this increased risk. Cardiomyocytes and fibroblasts express Toll-like receptor 4 (TLR4); long-chain saturated fatty acids can activate TLR4 by forming a ternary complex with it and the plasma protein fetuin-A, and palmitate has been shown to activate TLR4 on cardiomyocytes in vitro [458,459,460,461,462]. This receptor likewise responds to lipopolysaccharides and certain proteins signaling inflammatory damage, mediating the adverse effect of chronic infections (such as periodontitis [463,464,465]) on VH/HF risk and of sepsis on cardiac function [466,467,468,469,470,471]. TLR4 activation in cardiomyocytes can lead to oxidative stress, CaMKII activation, ER stress, and apoptosis [461,472,473,474]. Increased synthesis of diacylglycerol and of ceramide may also mediate adverse effects of saturated fatty acids on cardiomyocytes [475,476,477]. A recent prospective study of patients with HF found that saturated fat intake correlated positively with mortality rate, whereas polyunsaturated intake correlated negatively [478]. Another prospective study has linked increased plasma levels of saturated fats (especially myristate) to increased risk for HF [479]. These findings suggest that diets in which long-chain saturated fats constitute a small fraction of total fat content might be relatively protective with respect to VH/HF, especially in those with metabolic syndrome, obesity or type 2 diabetes.

Plant-based diets tend to be relatively low in saturated fat as a percentage of total fat, and the studies of Esselstyn have shown that strict adherence to a whole-food low-fat plant-based diet, coupled with sufficient medication as needed to maintain optimal LDL cholesterol levels, can achieve almost complete prevention of cardiovascular events and mortality in patients with advanced coronary disease, including those with previous MIs [480,481,482]. While these measures have been found to stop or reverse progression of coronary atherosclerosis, its seems likely that this low mortality also reflects a favorable impact on risk for, or progression of, VH/HF, that is independent of improved coronary perfusion. 

While whole-food low-fat plant-based diets tend to be exceedingly low in saturated fat, they are also relatively low in bioavailable phosphate, owing to the fact that much of their phosphate content is tied up in phytates that are poorly digestible [483,484]. Diets high in bioavailable phosphate promote calcification of arterial media as well as heart valves—factors which can increase VH risk by boosting afterload [485,486,487,488]. Additionally, high intakes of absorbable phosphate prompt a compensatory increase in bone production of the phosphaturic hormone FGF23 [489,490]. FGF23 is one of those hormones that, like angiotensin II or endothelin, can act on cardiomyocytes to activate PLC and thereby promote VH/HF [8]. Third-World populations consuming quasi-vegan diets low in phosphate additives are characterized by plasma FGF23 levels remarkably lower than those seen in Western populations [491,492]. Since phosphate additives are widely used in convenience foods, it is desirable to avoid these in order to moderate dietary phosphate absorption [493,494].

Plant-based diets that are relatively low in protein may also protect the heart by boosting plasma levels of fibroblast growth factor 12 (FGF21). A recent study found that, when a diet providing 8% of calories was fed in ad libitum amounts for 6 weeks to human volunteers, plasma FGF21 levels doubled; this effect may reflect hepatic activation of the kinase GCN2, which is activated by relative deficiencies of essential amino acids [495,496]. When obese subjects (BMI > 35) were excluded, the plasma FGF21 levels of long-term vegans were found to be about 3-fold higher than those of omnivores [497]. GCN2 promotes transcription of the FGF21 gene by increasing expression of the ATF4 transcription factor [498]. Recent studies show that cardiomyocytes are responsive to the hormone FGF21 via FGFR1/β-klotho receptors, and that this hormone is protective in rodent models of VH/HF [499,500,501,502,503]. Moreover, transgenic mice which overexpress FGF21 enjoy an extension of maximal lifespan comparable to that seen in mice subjected to marked caloric restriction—hence FGF21 has been referred to as “the longevity hormone” [504,505,506]. Whereas the protein content of most Western diets is considerably higher than 8%, the Okinawan people, while following their traditional quasi-vegan diet back in the 1950s, consumed about 9% protein calories—and were subsequently characterized by the highest proportion of centenarians in the world [507]. A wholly plant-based diet that provides a significant proportion of its calories from low-protein foods such as fruit, tubers, olives, avocado, and oils, and that is not unduly high in legumes and soy products can be expected to be sufficiently low in protein to promote FGF21 production. Moreover, plant protein, as opposed to animal protein, tends to be relatively low in certain essential amino acids (notably methionine and lysine); hence, a low-protein diet featuring plant protein can be expected to activate GCN2 more effectively than a diet of comparable protein content featuring animal protein [508]. 

GCN2, like the ER stress-activated kinase PERK, is a kinase for eIF2α [509]. Through this mechanism, both GCN2 and PERK help to control ER stress by slowing the synthesis of many proteins, while increasing the transcription of others that are protective in this regard [510]. However, they also boost expression of CHOP, a protein which is pro-apoptotic [213]. The net impact of eIF2α phosphorylation on VH/HF is the subject of conflicting reports; activation of GCN2 with the drug halofuginone, or up-regulation of eIF2α phosphorylation with the phosphatase inhibitor salubrinal, have been found to protect the heart from pressure overload [511,512,513,514,515]—whereas mice with genetic knockout of GCN2 have also been reported to be protected in this regard [516]. Hence, whereas moderate protein restriction might be expected to benefit VH/HF via increased FGF21 activity, the effect of directly activating GCN2 in cardiomyocytes is more equivocal. Animal studies are needed to test the impact of moderate protein restriction in rodent models of VH/HF.

Long-term vegans tend to be notably leaner than either vegetarians or omnivores; a thermogenic effect of FGF21 may be partially responsible for this finding [495,496,517,518,519,520]. Additionally, independent of body weight, vegans are at notably lower risk for type two diabetes [521,522]. Hence, a plant-based diet may have a favorable impact on two key risk factors for VH/HF—obesity and diabetes.

It appears likely that a low-fat plant-based diet can favorably influence flow-mediated vasodilation [523,524,525,526]. This may help to explain the fairly rapid impact of such a diet on anginal symptoms in patients with coronary stenosis (regression of stenotic lesions is only seen after many months of such a diet), and would be expected to provide symptomatic benefit in HF. 

Although the impact of long-term consumption of plant-based diets per se on risk for HF has not been assessed to-date, prospective studies have observed that a Mediterranean-style diet is protective in this regard, that whole grains are associated with protection whereas red meat, egg and dairy product consumption correlates with increased risk, and that high plasma vitamin C levels—likely serving as a marker for fruit and vegetable intake—predict decreased risk for HF [527,528,529,530,531]. 

Moreover, the first case report linking adoption of a wholly plant-based diet with improved control of HF has appeared [532]. The 79-year-old patient presented with 3-vessel coronary disease, aortic regurgitation, a 35% ejection fraction, and dyspnea upon exertion; he was already medicated with atenolol, candesartan, and low-dose aspirin. Declining coronary bypass and valve replacement therapy, he adopted a plant-based diet that excluded all animal products. After two months on this diet, in addition to an 18 pound weight loss and marked improvements in serum lipids, his dyspnea was eased to the point that he could adopt a regular aerobic exercise program. At the 3.5 month point, an echocardiogram revealed that his ejection fraction had increased to 50%. A formal trial of this diet-based strategy for management of HF may be warranted. 

Wholly plant-based diets require supplementation with vitamin B12, as is well known. Such diets are also devoid of carnitine and taurine, two factors potentially beneficial for the failing heart. Whereas humans can synthesize these factors themselves, their tissue levels in vegans tend to be somewhat lower than in omnivores [533,534]. Hence, carnitine and taurine supplementation may be particularly recommendable in VH/HF patients following a strictly plant-based diet.

In individuals who are salt-sensitive, salty diets provoke an increase in intravascular volume that is compensated by increased adrenal production of the natriuretic factor marinobufagenin (MBG) [535]. Curiously, this hormone (also produced in the sweat glands of toxic toads—hence its name!) can interact with plasma membrane sodium-potassium ATPases (a.k.a. “the sodium pump”) which express alpha1 subunits, both to inhibit pump action, and to induce intracellular signaling [536,537,538]. In kidney tubules, MBG exerts a compensatory natriuretic effect, presumably reflecting the role of the sodium pump in sodium retention [539]. In vascular smooth muscle, it promotes vasoconstriction by increasing intracellular sodium; this in turn boosts intracellular free calcium by stimulating the plasma membrane sodium/calcium exchanger [536,540]. (The structurally homologous sodium pump inhibitor ouabain exerts its inotropic effect on cardiomyocytes via a similar mechanism [541]) This effect likely rationalizes the pro-hypertensive effects of salty diets on salt-sensitive people and rodents [542]. However, MBG can also exert a hormonal effect on other tissues via the sodium pump—in particular, MBG can act directly on cardiac fibroblasts to activate intracellular signaling that promotes their conversion to myofibroblasts and boosts their collagen production [537,543,544]. The possibility that MBG might also act on coronary endothelium to suppress its production of NO via a pro-oxidant effect, merits consideration [545]. Epidemiological studies demonstrate an increased risk for LVH in salt-sensitive subjects and in those eating relatively salty diets, independent of blood pressure; MBG appears likely to be a mediator of this effect [546,547,548,549]. Moderation in dietary salt intake may therefore help prevent VH/HF. It should be noted however, that severe salt restriction can exert a countervailing adverse effect on the heart by boosting sympathetic and renin-angiotensin activity; that might explain why correlations between dietary sodium intake and mortality show a U-shaped pattern in some studies [550,551]. This also explains why drugs antagonizing angiotensin II activity increase the utility of dietary sodium reduction for hypertension control [552]. 

With respect to potassium, it is known that increased dietary potassium can exert a natriuretic effect in the context of salty diets; it is logical to suspect that this effect could decrease MBG production, although this possibility has not been evaluated [553]. In the Strong Heart Study, the sodium/potassium ratio correlated significantly with left ventricular mass; an inverse correlation with potassium per se did not achieve statistical significance [554]. In overview, moderation in the use of added salt—perhaps abetted by the use of modified salts that are potassium-enriched [555,556]—coupled with an ample intake of potassium-rich natural foods, appears to be a wise policy for prevention/control of VH/HF and for protection of cardiovascular health more generally. 

It is well documented that regular exercise training can provide symptomatic benefit in HF [557,558,559]. This is at least partially attributable to the favorable impact of episodes of increased shear stress on the function of vascular endothelium, promoting efficient flow-mediated vasodilation; this helps to compensate for the adverse impact of chronically decreased cardiac output on endothelial function [560,561,562]. Intriguingly, sauna can exert a comparable effect on vascular endothelium, and regular sauna treatments have been shown to benefit HF patients [75,563,564,565,566,567,568]. Sauna may be a good alternative for patients in whom significant aerobic exercise is too taxing; however, it can also be used in conjunction with exercise training to achieve better results [566]. Perhaps it should not surprise us that, in the general Finnish population, regular sauna bathing is associated with decreased overall and cardiovascular mortality—paralleling the effects of regular exercise [569].

## 8. Summing Up

To review in brief the measures discussed here which appear to merit consideration for inclusion in nutraceutical/dietary/lifestyle programs for preventing and controlling VH/HF: 

PhyCB could be employed to suppress oxidant stress stemming from Nox2- and Nox4-dependent NADPH oxidase complexes. CoQ may decrease oxidative stress of mitochondrial origin by accepting electrons from complex I—the chief mitochondrial site of superoxide production in VH/HF—and by its direct scavenging activity; it might also aid the efficiency of ATP production by cardiomyocyte mitochondria. Astaxanthin may limit mitochondrial superoxide generation by protecting the mitochondrial inner membrane and respiratory chain from cumulative oxidant damage. NAC supplementation should boost cellular levels of glutathione, which acts in various way to counteract the impact of hydrogen peroxide on cellular signaling. Phase 2 inducers such as lipoic acid or the widely distributed phytochemical ferulic acid should likewise be useful in this regard, while inducing increased expression of an array of antioxidant enzymes. Melatonin can up-regulate phase 2 induction by boosting Nrf2 expression, while also opposing TGFβ-mediated cardiac fibrosis by induction of Sirt1. Zinc may provide antioxidant protection via induction of metallothionein—a natural antagonist of cadmium toxicity—and possibly methionine sulfoxide reductase-A as well. Correcting selenium deficiency should ensure adequate expression of protective peroxidase and thioreductase activities. 

While control of oxidative stress will blunt a key stimulant of ER stress, the latter can also be addressed with TUDCA, which aids efficient and proper protein folding. Berberine, an activator of AMPK, can also combat ER stress, by slowing the rate of protein synthesis via down-regulation of eEF2 activity. 

Promoting recoupling of NOS should be beneficial both for supporting effective NO synthesis, and for quelling superoxide production. This may be achieved with supplemental citrulline—to offset the inhibitory/uncoupling activity of ADMA—and possibly high-dose folate, which may boost expression of dihydrofolate reductase to promote reduction of oxidized tetrahydrobiopterin, and may also protect tetrahydrobiopterin from oxidation. High-dose biotin may mimic the signaling impact of NO via direct stimulation of soluble guanylate cyclase. Soy isoflavones, via activation of ERβ, should boost expression and activity of eNOS in cardiac tissue. Cocoa flavanols and hawthorn OPCs up-regulate eNOS activity in vascular endothelium, thereby improving endothelium-dependent vasodilation and decreasing left ventricular afterload. Production of protective H_2_S could be promoted by supplemental NAC, and recent evidence also suggests that taurine also might have utility in that regard. In any case, taurine exerts a worthwhile positive inotropic impact on hear function in HF that appears to be safe.

An ample intake of EPA, particularly in the context of a low-fat diet, may oppose cardiac fibrosis through activation of fibroblast GPR120 receptors. Achieving an adequate membrane content of EPA/DHA, and a high ratio of EPA/AA in membranes and plasma lipids, can also be expected to protect VH/HF patients by decreasing their elevated risk for sudden death arrthymias and stabilizing their coronary plaque. 

Insuring that Mg levels in cardiomyocytes are in the high-normal range may curb to a modest degree the elevated calcium-mediated signaling in VH/HF by competing with free calcium for binding to calmodulin and possibly other calcium-activated proteins. The orotate salt of Mg has shown particular clinical activity in HF, possibly because orotate supplementation boosts cardiomyocyte levels of carnosine, which has antioxidant, membrane protective, and acid-buffering properties. The utility of carnitine supplementation in VH/HF may also reflect control of intracellular acidity, as, in the context of relative hypoxia, carnitine acts to disinhibit pyruvate dehydrogenase, directing more pyruvate to oxidation as opposed to lactic acid generation. Via activation of plasma membrane chloride channels, supplemental glycine may moderate the calcium influx through myocardial L-type calcium channels that plays a key role in the over-activation of NFAT and CaMKII driving the VH/HF syndrome. Copper supplementation may support effective myocardial angiogenesis, helping to ward off the microvascular rarefication that that plays a key role in the transition of VH to HF. 

A plant-based diet low in saturated fats may diminish the heart’s exposure to saturated fatty acids, thereby lessening the adverse impact of metabolic syndrome/obesity/diabetes on progression of VH/HF. Such as diet—particularly if phosphate food additives are concurrently avoided—should also lessen the adverse impact of FGF23 on the heart by moderating phosphate absorption. Conversely, a plant-based diet of moderate protein content could be expected to protect the heart by enhancing FGF21 production. While moderate protein restriction might also lessen myocardial ER stress by activating GCN2, this could also be expected to increase production of the pro-apoptotic transcription factor CHOP; hence, the direct impact of protein restriction on the heart in VH/HF requires further study. In people who are salt-sensitive, moderating dietary salt intake should protect the heart by decreasing plasma levels of MBG, a natriuretic factor that promotes cardiac fibrosis. Whether increased dietary potassium might blunt the adverse impact of sodium chloride in this regard, merits consideration. HF is characterized by impaired endothelial function, since decreased cardiac output lessens protective shear stress; this in turn increases the heart’s workload. Regular exercise training or sauna bathing may help to correct this problem by exposing the vasculature to episodic increases in shear stress.

Many of the measures discussed here are recommended on theoretical grounds, including rodent studies. However, it should be noted that CoQ, berberine, citrulline, taurine, Mg orotate, exercise, and sauna have received greater or lesser degrees of clinical confirmation in controlled studies targeting HF patients. Table 1 provides rough “guesstimates” of the clinically useful doses of various nutraceuticals that have potential for VH/HF control. The dose ranges provided have been found to exert physiological effects in clinical research. 

There is evidently considerable scope for the development of functional foods and complex supplements which could make it more convenient and less expensive to ingest a number of nutraceuticals of potential benefit in VH/HF. Such supplements would not be expected to contain all of the agents discusses here, as there evidently will be a point of diminishing returns with respect to the net benefit achieved by addressing a specific target—i.e., a person who is employing 2 or 3 agents that boost NO bioactivity might derive little further benefit from expanding that list to 5. The strategy of employing multiple nutraceuticals for management of HF has been suggested previously by other analysts [598,599,600,601]. Indeed, a supplementation regimen providing CoQ10, taurine, and carnitine has been shown to decrease LV end diastolic volume in patients with diminished ejection fraction awaiting coronary bypass surgery [600]. The beneficial effects of multi-vitamin-mineral supplementation on LV function in HF patients were noted above [430]. Table 2 suggests a way to configure the suggested nutraceuticals into a practical program of supplementation.

It should be noted that, whereas the agents and measures discussed in this essay have targeted the VH/HF syndrome, some of them likely would also work to promote vascular health in other, complementary ways: opposing atherogenesis, stabilizing plaque, preventing vascular calcification and aneurysms, moderating systemic and pulmonary blood pressure, and controlling platelet aggregation. This reflects the fact that oxidative and ER stress, insufficient NO or H_2_S bioactivity, and magnesium deficit contribute in various ways to these vascular pathologies [376,584,602,603,604,605,606,607,608,609,610,611,612,613,614,615,616,617,618,619,620,621,622,623,624,625,626,627,628,629,630,631,632,633,634,635,636]. Moreover, these factors also contribute to the adipocyte dysfunction—hypertrophy, insulin resistance, elevated production of pro-inflammatory adipokines and diminished production of adiponectin—that underlie the metabolic syndrome and diabetes, key drivers of vascular disease [637,638,639,640,641,642,643,644,645,646,647,648,649,650,651,652,653,654,655,656,657,658,659,660]. Additionally, glycine, FGF21-boosting plant-based diets, and aerobic exercise training can be expected to favorably influence adipocyte function and insulin sensitivity [649,661,662,663,664,665,666,667]. With respect to berberine, it also acts to lower LDL cholesterol—through a mechanism that is complementary to that of statins [668,669]—and can be used in combination with monacolin-rich red yeast rice as a well-tolerated nutraceutical alternative to prescription statins [670].

## Figures and Tables

**Table 1 ijms-22-03321-t001:** Nutraceuticals for Prevention and Control of VH/HF—Suggested Daily Doses.

Phycocyanobilin.	100 mg (or 15 g spirulina) [140,570].
N-Acetylcysteine	1200–1800 mg [152].
Lipoic Acid	1200–1800 mg and/or Ferulic Acid 250–1000 mg [571,572,573].
Zinc	30–80 mg (complemented by copper, 1–2 mg) [184,574].
Selenium	100–200 mcg [189,193].
Ubiquinol	300 mg [575,576].
Astaxanthin	10–20 mg [577,578].
Melatonin	5–20 mg (at bedtime) [579,580].
TUDCA	2–4 g [581].
Berberine	1000–2000 mg [582].
EPA	1–2 g daily [583].
Magnesium	200–400 mg [584].
Mg Orotate	3–4 g [390].
Citrulline	3 g [585,586].
Folate	40–80 mg [247].
Na or K Nitrate	500–1000 mg (or 250 mL beet juice) [587,588].
Biotin	20–40 mg [589,590].
Soy Isoflavones	100 mg [591,592].
Cocoa flavanols	400–1000 mg [593,594].
Taurine	2–6 g [407,415].
Carnitine	2–4 g [399].
Glycine	10–15 g [595,596].
Copper	2–8 mg [431,597].

**Table 2 ijms-22-03321-t002:** Practical Guidance for Heart-Protective Supplementation.

**Nutraceutical Regimen Suggested for VH/HF Prevention**
Multivitamin/mineral—Includes Mg, Zn, Cu, Se
Drink Powder—Spirulina, Citrulline, Taurine, Glycine, Soy Isoflavones, Cocoa Flavanols
Glutathione Booster Caps—N-Acetylcysteine, Lipoic Acid, Ferulic Acid
EPA/DHA Caps
Melatonin Cap
**Nutraceuticals Which Could be Added for VH/HF Treatment**
ER Stress Caps—Berberine, TUDCA
Ubiquinol
Carnitine
High-dose Folate
High-dose Biotin
**Consider Also:**
K Nitrate caps (or beet juice)
Mg Orotate
Astaxanthin

Ingredients chosen for the Drink Powder—excepting spirulina—are characterized by mild or absent flavor and high solubility. Once PhyCB becomes available in capsule form, this could be provided separately, such that spirulina would no longer need to be included in the Drink Powder. The “glutathione booster” product would be of particular value for the elderly, as Nrf2 activity and glutathione levels decline with increasing age.

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
