# Peer review of "Nutraceutical, Dietary, and Lifestyle Options for Prevention and Treatment of Ventricular Hypertrophy and Heart Failure"

_ijms, 2021, doi:10.3390/ijms22073321_

Round 1
Reviewer 1 Report
Review is focused on an actual and important topic. I have some questions and comments:
1. In Table 1 are presented suggested daily doses of nutraceuticals for prevention and control of VH/HF. I assume that the presented doses represent summary of data obtained in several studies. The corresponding references for presented data should be included into the Table.
Minor comments:
In review you mentioned several times Nrf2. The abbreviation and function of this protein could be explained more in detail.
Please correct nrf2 to Nrf2.
Check in text the formula of Hydrogen sulfide – sometimes is H2S and not H2S.
Author Response
As requested, I have added citations to each of the cited nutraceuticals in Table 1 to provide the reader with insight regarding the basis of the dose ranges I suggest.
I have added text to page 4 (highlighted in yellow) which explains the meaning and function of Nrf2. I have also corrected the mentions of Nrf2 and H2S throughout the manuscripts.
Many thanks for your careful attention to this lengthy submission.
Reviewer 2 Report
The review article "Nutraceutical, Dietary, and Lifestyle Options for Prevention and Treatment of Ventricular Hypertrophy and Heart Failure" summarized the existing evidence on the effects of nutraceuticals, diet and lifestyle on left ventricular hypertrophy and heart failure
The paper is well organized and written
Minor points
1) Please add a research strategy of the studies selected for the review
2) Table 2 should be modified as follow
Nutrauceticals for VH/HF prevention .. .. .. Nutrauceticals for VH/HF "treatment" .. .. ..Author Response
At the end of the abstract, I have added a sentence (highlighted in yellow) that explains my use of PubMed as the source of the research papers whose findings I cite.
I have revised the titles on Table 2 as suggested.
Many thanks for making the effort to review this lengthy manuscript.